# *Kosteletzkya pentacarpos*: A Potential Halophyte Candidate for Phytoremediation in the Meta(loid)s Polluted Saline Soils

**DOI:** 10.3390/plants10112495

**Published:** 2021-11-18

**Authors:** Mingxi Zhou, Stanley Lutts, Ruiming Han

**Affiliations:** 1Biology Centre, Czech Academy of Sciences, Institute of Plant Molecular Biology, 37005 Ceske Budejovice, Czech Republic; mingxi.zhou@umbr.cas.cz; 2Groupe de Recherche en Physiologie Végétale (GRPV), Earth and Life Institute-Agronomy (ELIA), Université Catholique de Louvain, 5 (Bte 7.07.13) Place Croix du Sud, 1348 Louvain-la-Neuve, Belgium; stanley.lutts@uclouvain.be; 3School of Environment, Nanjing Normal University, Nanjing 210023, China; 4Jiangsu Center for Collaborative Innovation in Geographical Information Resource Development and Application, Nanjing 210023, China

**Keywords:** *K. pentacarpos*, meta(loid)s, cadmium, zinc, salinity

## Abstract

*Kosteletzkya pentacarpos* (L.) Ledebour is a perennial facultative halophyte species from the Malvacea family that grows in coastal areas with high amounts of salt. The tolerance of *K. pentacarpos* to the high concentration of salt (0.5–1.5% salinity range of coastal saline land) has been widely studied for decades. Nowadays, with the dramatic development of the economy and urbanization, in addition to the salt, the accumulation of mate(loid)s in coastal soil is increasing, which is threatening the survival of halophyte species as well as the balance of wetland ecosystems. Recently, the capacity of *K. pentacarpos* to cope with either single heavy metal stress or a combination of multiple meta(loid) toxicities was studied. Hence, this review focused on summarizing the physiological and biochemical behaviors of *K. pentacarpos* that has been simultaneously exposed to the combination of several meta(loid) toxicities. How the salt accumulated by *K. pentacarpos* impacts the response to meta(loid) stress was discussed. We conclude that as a potential candidate for phytoremediation, *K. pentacarpos* was able to cope with various environmental constrains such as multiple meta(loid) stresses due to its relative tolerance to meta(loid) toxicity.

## 1. Introduction

In the world, approximately 830 million hectares of land, accounting for more than 10% of the world’s total land area, are affected by salinity [1]. Coastal land is one of the most important parts of salt-affected areas. The development of economic growth and urbanization in recent decades, has resulted in the excessive release of meta(loid)s to the coastal waters and land, which has caused severe contamination in these soils [2]. In meta(loid)-polluted environmental settings, several meta(loid)s often coexist and interact with each other in a complex way [3,4]. This is especially the case for Cd and Zn, which have similar chemical properties that are based on the same outer shell electron configuration, and they are frequently simultaneously present in polluted soils. Meta(loid) toxicity greatly impairs plant growth and development and thus induces a reduction in the biomass accumulation and grain yield of cultivated crops [5,6,7]. Furthermore, meta(loid) uptake by crops is especially risky due to the toxicity of meta(loid)s to human beings [8,9].

*Kosteletzkya pentacarpos* (L.) Ledebour (syn. = *Kosteletzkya virginica* (L.) Presl. ex Gray) is one species belong to a group of salt-dilution halophytes and is more commonly known as “seashore mallow” and “saltmarsh mallow”. It was introduced to China for economic and ecological purposes in 1993 [10,11]. In the past few decades, it has been proven that *K. pentacarpos* is able to grow in the coastal zones (NaCl concentration: 5–10‰) of Jiangsu, Liaoning, Tianjin, Shandong, Zhejiang, and Fujian provinces, which are located from the north (23°30′ N) to the south (38°45′ N) of China [12]. It has already been regarded as the main halophyte species that can be used as a tool in coastal saline ecological engineering to restore saline soil due to its high biomass, carbon fixation potential, and production of abundant bioactivators, such as polysaccharides, flavonoids, and saponins [12].

Recently, *K. pentacarpos* was proven to have capacities making it able to cope with multiple meta(loid) contamination in salty soil [13,14,15]. Hence, this review summarized the current understanding of the behavior of *K. pentacarpos* that has been exposed to NaCl and multiple meta(loid) pollutants simultaneously. In Section 2 and Section 3, the review focuses on the combination of Cd and Zn, uncovering how they interact. In Section 4, how *K. pentacarpos* behaves as a potential candidate for phytoremediation in saline soil contaminated with multiple meta(loid)s is discussed.

## 2. The Specific Response in *K. pentacarpos* Exposed to Simultaneously Cd and Zn Toxicity

### 2.1. The Antagonistic Relationship between Excessive Cd and Zn in K. pentacarpos

The physiological plant responses that take place during the photosynthesis [16], transpiration [17], nutrition absorption [18], and oxidative status directly depend on heavy metal absorption, translocation, and the eventual accumulation of these heavy metals in the plant. It is known that heavy metal toxicity is the consequence of internal accumulated elements, while external ions are not directly toxic by themselves, although they might indirectly influence the plant through impacts on the soil ecosystem [19] or through modifications in the soil structure [20]. Hence, interactions between the studied elements in terms of absorption by the roots and distribution within the plant is a crucial aspect.

In recent years, the impact of Cd on Zn uptake (and vice versa) in plants was widely studied [21,22], but until now, there has been no clear conclusion as to the relationship between Cd and Zn when they simultaneously exist in substrate. When considering absorption by *Ricinus communis* L., the interaction pattern of Cd and Zn showed antagonistic behavior with low concentrations of Cd and Zn (Cd: 1 mg kg^−1^ soil, Zn: 100 mg kg^−1^ dry weight soil), but they exhibited a synergistic behavior with high concentrations of the combination of Cd and Zn (Cd: 40 mg kg^−1^ soil, Zn: 800 mg kg^−1^ dry weight soil) [23]. In *K. pentacarpos*, regardless of whether it was exposed to Cd and Zn toxicity for 2 days or for 3 weeks, the excessive Zn had a negative effect on Cd accumulation in the root of plants that were simultaneously exposed to both heavy metals [14,15]. It is well established that Cd and Zn share numerous biochemical properties and that Cd is mainly absorbed by poorly selective Zn transporters (e.g., natural resistance-associated macrophage protein, NRAMP) [24,25]. An excess of Zn would therefore logically lead to a decrease in Cd absorption.

It was found that in *K. pentacarpos*, most Cd was fixed in cell debris that was mainly contained in the cell walls (74%), while for Zn, it was equally accumulated in the cytoplasm (45%) and in the cell debris (50%) [14]. According to literature, cell walls play a vital role in Cd fixation, and this allows Cd to be kept away from the most important metabolic places within the cell [26]. Furthermore, hemicellullose is a major component of the cell wall. An increase in the hemicellulose content in the halophyte *Sesuvium portulacastrum* when it was exposed to Cd as well as an increase in the mucilage, which is mainly composed of pectin, were recorded [27]. In the stem of *K. pentacarpos*, the hemicellulose content increased when it was exposed to excessive Zn alone [28]. Hence, it could be hypothesized that the combination of Cd and Zn may also modify the structure of polysaccharides to accumulate heavy metals in *K. pentacarpos*, which should be tested for confirmation in the future.

The root sequestration of heavy metals is a strategy that can be used to avoid the accumulation of toxic ions in photosynthetic tissues. In *K. pentacarpos*, such a process has been clearly demonstrated, and heavy metal accumulation is always higher in the root than it is in the shoot [14]. In order to avoid a perturbation of the root metabolism, heavy metal accumulation cannot exceed a threshold level and requires detoxifying mechanisms such as phytochelatins (PC) oversynthesis and the vacuolar sequestration of the PC–metal complexes. *K. pentacarpos* is able to trigger such an adaptative response, as suggested by the recorded increase in root PC observed in metal-treated plants. However, the fact that PC content may be higher in the leaves than in the roots while the Cd and Zn concentration is lower in the former than in the latter leads us to hypothesize that metal sequestration by non-protein thiol may be, to some extent, overcome in the below-ground part of the plant. Besides root sequestration, stem retention can be regarded as a complementary strategy to reduce heavy metal accumulation in photosynthetic leaves. In another halophyte plant species (*Atriplex halimus*), it has been demonstrated that the precipitation of Cd in oxalate crystals specifically occur in the stem [29]. The presence of mucilage within the xylem vessels [30] may offer *K. pentacarpos* a convenient opportunity to retain heavy metals, as suggested by the recorded increase in uronic acid of the stem mucilage in Zn-treated plants [28]; however, interaction between Cd and Zn in the absorption site still remains an open question.

Previous data have suggested that Cd or Zn could be xylem-loaded by either the symplastic or apoplastic pathway in *K. pentacarpos* [31]. In dwarf polish wheat, it was demonstrated that several metal transporters such as *cadmium-transporting ATPase* and *plant cadmium resistance 4* were specifically regulated by a combination of Cd and Zn treatment [14]. It has also been determined that other metal transporters are able to transport both Cd and Zn, including Arabidopsis AtNRAMP3 and AtNRAMP4 [32]. Another group known as heavy metal ATPases has two subgroups that are based on their metal substrate specificity: A Cu/Ag group and a Zn/Co/Cd/Pb group [33,34]. Takahashi [35] suggested that OsHMA2 plays a role in Zn and Cd loading to the xylem and that it participates in the root-to-shoot translocation of these metals in rice.

Nowadays, many heavy metal transporters are recognized with the understanding of their structure and function [36,37,38]. In the future, studies should focus on: (1) when a halophyte plant species is simultaneously exposed to multiple heavy metals excess, how does the plant perceive and select the different heavy metal ions to uptake? (2) How does the plant distribute transporters to target different heavy metal ions to minimize the damage?

### 2.2. The Oxidative Status and Antioxidant System in K. pentacarpos

Recently, the oxidative stress and antioxidant system in *K. pentacarpos* was described in response to a combination of Cd and Zn toxicity. Although Cd does not directly participate in a redox reaction, such as the Fenton reaction, it is still capable of disturbing the electronic transmission chain during photosynthesis in *K. pentacarpos* [14]. A combination of 10 µM Cd and 100 µM Zn strongly inhibits the photosynthesis process, triggering nonphotochemical quenching (NPQ) to dissipate excessive energy [39]. It has been indicated that excessive Zn could exacerbate the negative effects of Cd, which induces a decrease of the variable chlorophyll fluorescence intensity ratio (F_690_/F_730_) and in the stress adaption index, suggesting a reduction of potential photosynthetic activity [40]. Hence, it could be hypothesized that Cd can affect the electronic transmission chain to inhibit the photosynthesis and to induce oxidative stress by generating reactive oxygen species.

Although Cd accumulation was reduced in roots and leaves in the combined treatment, oxidative stress was clearly induced, as indicated by the generation of malondialdehyde (MDA), carbonyl, and H_2_O_2_ [14]. However, the interaction between Cd and Zn in terms of oxidative stress may depend on the considered parameters. For example, no difference was found between Cd alone and Cd + Zn treatment for the relative leakage ratio, which indicates that the membrane was mainly damaged by Cd, resulting in an outflow of ions [41]. Nevertheless, MDA increased in response to Zn alone [41]. However, it was lower in the combination Cd and Zn treatment, which indicated that the biological membrane may be affected by Zn but that this deleterious impact was somewhat masked by the toxicity of Cd in response to mixed treatment.

To cope with the oxidative stress induced by mixed toxicity, high global antioxidant activity was described in *K. pentacarpos*. The accumulation of reduced glutathione (GSH) in the root and its depletion in the leaf was recorded in the mixed treatment. It was found that on a short-term basis (2 h) in *Arabidopsis*, the root GSH was preferentially allocated to the synthesis of the PC involved in Cd chelation, which led to decreased GSH levels, without any alternative pathways being activated to complement GSH’s antioxidative functions [42]. After one day of adjustment, multiple antioxidative pathways increased, including the ascorbate–glutathione cycle [42]. In leaves of *K. pentacarpos*, the depletion of the reduced GSH could mainly be attributed to the synthesis of phytochelatins [14]. It can be concluded that *K. pentacarpos* tries to find a compromise to balance the glutathione activity as a free radical scavenger and the synthesis of PC as an efficient metal chelator under the combined toxicity of Cd and Zn. However, there is still no experimental proof that PC are equally involved in Cd and in Zn chelation. Although it is sometimes considered that Cd chelation by a non-protein thiol is more efficient than Zn chelation, it could also depend on the type of synthesized PC [14]. In addition to non-enzymatic antioxidants, enzymatic antioxidants such as SOD, CAT, etc., also have important functions for detoxification under heavy metal stress [43,44]. Han et al. [43] indicated that the excessive Cd alone enhanced dehydroascorbate reductase, glutathione reductase, and peroxidase activity to detoxify ROS induced by Cd exposure. However, when plants are under multiple heavy metals stress, how specified enzymatic antioxidants act is still unknown. Moreover, the antioxidant system status in plants differs from heavy metal exposure duration. The mechanism regarding the dynamic changes of the antioxidant system with plant acclimation to the stress of multiple heavy metals should be considered more deeply in the future.

### 2.3. Plant Hormones Status and Osmoprotectants in K. pentacarpos

When plants suffer from abiotic stress, phytohormones play vital roles in stress perception, signal transduction, and stress resistance [45,46,47]. Besides the “classical” phytohormones (e.g., auxin, cytokinin (CK), and ethylene), polyamines (PAs), mainly putrescine (Put), spermidine (Spd), and spermine (Spm), play critical and multiple functions in plants, such as free radical scavenging [48], the stabilization of the biological membrane, the regulation of mineral nutrition, and the regulation of the cell cycle [49], and are also involved in abiotic stress signaling [50]. Starting from arginine, two pathways are known to form Put via ornithine by arginase and ornithine decarboxylase and via arginine through arginine decarboxylase. Additionally, ornithine is also an important substrate for proline biosynthesis [51]. Spd and Spm are synthesized from Put associated with S’adenosyl-l-methionine (SAM) [52]. Furthermore, SAM is the shared precursor in ethylene and Spd as well as in Spm synthesis. Recently, the modification of the polyamines as well as the ethylene status induced by Cd and Zn was pinpointed [41].

In *K. pentacarpos*, the concentration of Put was increased, and the concentration of Spd was reduced in Cd alone treatment and mixed treatment, while they were not modified by Zn alone treatment [41]. It has been indicated that Cd can induce polyamines modification, including when *K. pentacarpos* is exposed to a combination of Cd and Zn. Furthermore, the exogenous application of PAs indirectly indicates the capacity of PAs to enhance plant tolerance to heavy metals. Considering the multiple roles of PAs in plant metabolism [53], these approaches frequently remain descriptive, and the real underlying cause(s) of the recorded improvement remain elusive. It was found that exogenous spermidine could alleviate the accumulation of ROS (superoxide anion, hydrogen peroxide, malondialdehyde) in relation to an increase in the soluble proteins and antioxidants in *S. matsudana* leaves under the corresponding cadmium stress [54]. Meanwhile, the increased concentration of Put was accompanied by a concomitant increase in the ethylene concentration in the leaves in *K. pentacarpos* [41]; consequently, this led to a decrease in the SAM that was available for Spd and Spm synthesis. Hence, when testing this hypothesis in the future, the quantification of SAM is absolutely necessary. Moreover, there are a lot of puzzling issue between PAs and ethylene metabolism. This would be imperative to unveil biochemical as well as molecular insights into the combined ethylene and PAs-mediated control of metal-specific toxicity in plants.

Although excess Zn (100 μM in the nutrient solution) did not modify the polyamines and ethylene metabolism, it had a possible impact on other kinds of phytohormones. The CK-dependent regulatory module underlined the importance of the maintenance of zinc nutrition in rice [55]. These authors provided unequivocal evidence that cytokinins have a key role in controlling the Zn status in plants. In *K. pentacarpos*, the excessive Zn drastically reduced CK concentration, but Cd had no impact on it. What is interesting is that excessive Zn strongly decreased the total cytokinin content in the plant leaves, even in combination of Cd and Zn treatment. According to Atici et al. [56] and Xu et al. [57], a high Zn concentration could have strong effects on CK metabolism and induced a decrease in the CK content in plants, while optimum Zn concentration increased their content. In addition to cytokinin, auxin, ABA, and SA as well as JA were more significantly affected in response to the mixed treatment than in the case of exposure to one single heavy metal [56,57]. It was suggested that the high concentrations of Zn (1.0 and 10 mM) decreased the contents of zeatin, zeatin riboside, and gibberellic acid in germinating chickpea seeds in relation to the enhanced ABA content [56]. Wu et al. [58] related plant hormone status to the ascorbate (AsA)GSH cycle and hypothesized that the 6-benzylaminopurine (BAP, one synthetic CK)-regulated AsA–GSH cycle is mediated by the CK signal pathway via IAA and ABA in the leaves of eggplant (*Solanum melongena* L.) seedlings under 10 mM zinc stress. For *K. pentacarpos*, the combined stress of Cd and Zn induced a specific physiological modification in the plant, and it seems highly probable that, for some phytohormones, Cd and Zn act on a common target but not necessarily in the same way and not necessarily at the same step of the biochemical pathway. The lack of information regarding the *K. pentacarpos* genome and the fact that mutants are consequently not available hamper a more sophisticated experimental approach.

In addition to phytohormones, widespread osmoprotectant proline exists in almost all kinds of plant species as an essential amino acid, while glycinebetaine (GB) accumulates in a limited number of plant species, such as halophyte plants [59]. The combination of Cd and Zn toxicities significantly reduced proline biosynthesis compared to Cd alone in *K. pentacarpos* [15]. Tripathi et al. [60] found that the activities of pyrroline-5-carboxylate synthetase (P5CS) and pyrroline-5-carboxylate reductase (P5CR) were increased immediately in *Triticum aestivum* (Wheat) exposed to excessive Cu and Cd. It remained higher through the end of the experiment. However, the ornithine amino transferase activity of the treated plants was lower than that of the P5CS and P5CR enzymes [60]. The proline dehydrogenase activity sharply decreased in the early phases of combined Cd and Zn exposure (up to 12 h) but remained unchanged thereafter until the end of the experiment. It has been suggested that the glutamate pathway and ornithine pathway play an equivalent role in proline synthesis in *K. pentacarpos* that has been exposed to the stress of heavy metal [15]. On the other hand, it is speculated that the shared precursor, ornithine, could also participate in Put synthesis under mixed treatments [15]. What is specific about the mixed treatment is that the concentration of Spd decreased in relation to an increased concentration of Spm compared to Cd or Zn alone [41]. This suggests that Spm synthesis from Spd may be specifically stimulated to cope with mixed toxicity since Spm harboring 4 positive charges at a cellular pH is often thought to be for the protection of cellular structures. After the aminoethoxyvinylglycine (AVG, ethylene synthesis inhibitor) was added, the accumulation of Cd was reduced in mixed treatment, which is associated with a decrease in Put concentration and a sharp increase in Spd and Spm [41]. For GB, the previous studies have shown that it mainly exists in the chloroplasts to protect the photosynthetic machinery [61]. It accumulates not only in leaves but also in the roots of *K. pentacarpos* [15]. Meanwhile, betaine aldehyde dehydrogenase (BADH) expression was activated in response to all of the treatments in the roots of *K. pentacarpos* [15]. Exogenous Cd strongly increased GB accumulation in the leaves, while Zn reduced it. The clear correlation between *KpBADH* gene expression and BADH activity suggests that Cd stress could induce GB synthesis so as to create a protective function [15]. Hence, it may be assumed that Cd and Zn not only differ in terms of their phytohormonal impact, but also for the type of osmoprotectant that they induce.

Until now, the involvement of several plant hormones and plant growth regulators has been found to be associated with heavy metal stress responses. However, the clear link between hormonal pathways and metal-binding ligands in plants, either due to certain signaling pathway or common synthesis pathway, still needs to be explained. Further investigations on hormone synthesis mutants or transgenic plants may help to identify clear interrelationships.

In summary, when *K. pentacarpos* is exposed to combination of excessive Cd and Zn, it actually has specific responses to the heavy metal toxicity during the photosynthesis and in the antioxidant system and phytohormones (Table 1). In the future, studies should focus on the molecular mechanism of plant tolerance to the toxicity of multiple heavy metals.

## 3. NaCl Improves the Plant Resistance to Combination of Cd and Zn Toxicities

### 3.1. NaCl Reduces Heavy Metal Accumulation and Improves Plant Antioxidant System in Mixed of Cd and Zn Treatment

Although it is well-known as a halophyte plant species, *K. pentacarpos* has been confirmed to be able to grow in a nutrient solution (1/2 strength Hoagland solution) without salinity [30]. Moderate doses of salinity may stimulate plant growth in halophyte plant species: this has been confirmed in *K. pentacarpos* exposed to 100 mM NaCl [30]. At this NaCl dose, the leaf water content remained unaffected, which to some extent, may be a consequence of leaf mucilage enhancement and modification in its composition. Hence, 50 mM NaCl was chosen to be added into the substrate, and it did not significantly improve plant growth in the absence of heavy metals, regardless of whether NaCl was added during acute (48 h) or long-term treatment (3 weeks) [15,41]. Growth stimulation should not be regarded as an absolute criterion for the definition of halophyte plant species on its own [53,66], and the plant physiology can be modified in a positive way that is independent of growth stimulation.

From a relative point of view, the recorded NaCl-induced decrease in Cd accumulation in the mixed treatment (74%) was higher than it was during treatment with Cd alone (44%) [14], which indicates that NaCl had a stronger impact on heavy metal absorption in *K. pentacarpos*, and especially higher impact on the absorption of Cd, during combined Cd and Zn treatment. It can be assumed that the accumulated Cd and Zn as well as the moderate accumulation of NaCl could interact and that NaCl could inhibit Cd translocation under conditions where Zn had accumulated [67]. Research should aim to confirm this in the future.

In addition reducing the accumulation of heavy metals, salinity also modifies the subcellular distribution of Cd [68]. In both roots and leaves of *K. pentacarpos* exposed to a combination of Cd and Zn in the presence and absence of salinity, Cd was mainly fixed in cell debris (more than 50%) [14], which was mostly composed of a cell wall. It was demonstrated that salt could induce increased cell wall thickening through the reinforcement of the secondary wall with hemicellulose, which improved the metal binding capacity [69]. It was noteworthy that when *K. pentacarpos* were stressed by Cd + Zn, the proportion of hemicellulose proportion increased significantly, which was one of the most important constituents for heavy metal sequestration outside of the cell [70]. In addition, NaCl increased the Cd proportion in the metal-rich granule (MRG) [71], which was probably related to the increase of cysteine-rich PC. The MRG is regarded as insoluble complexes with high molecular weights and is formed through the polymerization of low molecular weight of PC in vacuole [71]. It was indicated that the volume fraction of the vacuole was greater in the halophyte *Suaeda maritima* (L.) Dum grown under saline conditions when compared with those grown under non-saline conditions [72]. It could be hypothesized that NaCl stimulates a larger volume fraction of vacuoles allowing more storage of metal-binding PC.

Since the sulfur concentration increased in the stressed tissue of *K. pentacarpos*, it was also confirmed that *K. pentacarpos* triggered this PC protective mechanism, especially under Cd stress [14]. It also illustrated an extremely important implication that organ response to heavy metals was not only a matter of the total amount of accumulated ions but also directly depended on ion distribution. It still remains to be explained: (1) what is the signal transduction regulating heavy metal subcellular distribution? (2) How does salinity affect heavy metals distribution and complementation in molecular mechanism? The answers to these questions should be determined in the future.

The disturbance of the photosynthesis process induced by Cd and Zn accumulation directly leads to excessive electron generation [73]. NaCl increased all of the fluorescence-related parameters (except NPQ) and pigment concentration, suggesting that low doses of NaCl unexpectedly contributed to the protection of the chloroplast structure as well as of the photosynthesis process. Meanwhile, the direct consequence resulting from the improvement of photosynthesis was the reduced oxidative damage, which was indicated by the decreased concentration of carbonyl, H_2_O_2_, and MDA [14]. The additional NaCl also triggered non-enzymatic protection, and more especially, it increased the AsA and GSH concentration to alleviate heavy metal toxicity [43,74]. In addition, we also found that salinity addition significantly increased GB synthesis in *K. pentacarpos* exposed to Cd toxicity, which was thought to protect chloroplasts. It was reported that the higher accumulation of GB is able protect photosynthetic machinery under stress conditions [75].

### 3.2. NaCl Regulates Plant Hormones Status Allowing the Plant to Cope with a Combination of Cd and Zn Toxicities

In Radyukina’s study [76], the constitutive level of free Spd and Spm and the expression of their genes (*SPDS*, *SPMS*) in *Thelungiela halophila* were significantly higher compared to that of *Plantago major* L. Under a combination of excessive Cd and Zn in *K. pentacarpos*, the recorded concentration of ethylene was quite high, but salinity strongly reduced it. Ethylene is mainly involved in the senescence process in plants, and this implies that 50 mM of NaCl reduces senescence in the halophyte *K. pentacarpos* through a decrease in ethylene production [41]. This is an unusual behavior in the plant kingdom, and additional work must be undertaken to explain the impact of NaCl on ethylene synthesis. For PAs, with the decrease of ethylene, the concentration of Put was expected to decrease, but this was obviously not the case. We can assume that combination treatment activates another pathway (from ornithine) for Put biosynthesis in *K. pentacarpos*, which could be related to the proline synthesis [15].

In addition to ethylene and PAs, in the absence of exogenous trans-zeatin riboside, salinity significantly increased total cytokinin and the total gibberellins in *K. pentacarpos*. Numerous studies in the literature have introduced the effect of exogenous cytokinin as well as salinity on plant behavior [77,78,79]: the application of exogenous kinetin reduced the inhibitory effects of NaCl on K^+^ and Ca^2+^ uptake, improved antioxidant system assess through the determination of the ascorbate–glutathione cycle, and reduced oxidative damage in *Solanum lycopersicum* [80]; exogenous cytokinins (CPPU and 6-BA) also increased the photosynthesis process and reduced ROS generation that had been induced by salinity [81]. However, in all of these studies, the additional salinity was regarded as a stressed factor for the plant. The amount of 50 mM NaCl is likely a positive stimulus. However, the exogenous *t*-zeatin riboside assumes a key function in heavy metal resistance, especially in Zn-treated plants, but its efficiency was lower in the presence of NaCl in our halophyte *K. pentacarpos*, which could be attributed to a fact that the heavy metal reduced the endogenous CK concentration, while NaCl increased it.

## 4. Candidate for Phytoremediation of Polymetal-Contaminated Salt Areas

### 4.1. Exogenous NaCl in Nutrient Solution Compromises the Use of K. pentacarpos for Phytoextraction and Meta(loid)s Removal

Phytoremediation is an attractive alternative to expensive industrial approaches for decontamination but requires the use of meta(loid)-tolerant plants [26,82]. Hyperaccumulators are not suitable for meta(loid) removal since they usually produce a very low amount of biomass. Moreover, cleaning salt-affected sites that have been contaminated by meta(loid)s requires the use of a halophyte plant species. Hence, the selection of proper halophyte species for phytoremediation is a crucial issue nowadays.

It has been confirmed that halophyte species *K. pentacarpos* has the ability to accumulate various heavy metals (Zn, Cd, and Cu) in the absence of salinity [13,14,41]. Although the accumulation level was far below the level fixed for hyperaccumulating plants, its high amounts of biomass compensated for the consequences of this medium concentration in terms of the quantity of the pollutants removed by the plant. In optimal conditions, the plants may produce more than 30 tonnes dry matter per hectare per year [12].

However, without the consideration of meta(loid) bioavailability (under nutrient solution), moderate doses of salinity (50 mM) had a negative impact on meta(loid) absorption and accumulation, protecting the plant from ionic toxicity. It compromises the use of *K. pentacarpos* for phytoremediation [14,64]. The same results have been confirmed: when using a combination of Cd and Zn, the Cd concentration decreased even more strongly than it did in the treatment with Cd alone [14]. However, it must be considered that salinity alleviates oxidative damage that has been induced by excessive heavy metals through the activation of the antioxidant system [14] as well as through the modification of different plant hormones statuses [41,65], resulting in the improvement of metal resistance and plant growth, the extension of life span, and eventually, the enhancement of the total amount of toxic ion accumulation[62,63]. The impact of the treatment on the survival rate was not studied since the plants were exposed to heavy metals at a young age for limited periods ranging from 2 days to 2 weeks. It could be argued that this duration was not sufficient to fully apprehend all the long-term consequences of meta(loid) accumulation. Hence, the idea that *K. pentacarpos* that are still alive after 2 weeks could encounter problems after a longer stress exposure time cannot be excluded. The beneficial impact of NaCl would then be more useful for a phytoextracting approach if were to help the plant remain alive and grow for a longer period. It could thus be of primary importance to consider longer term experiments with this type of material to clearly identify the plant response at each phenological stage [66].

### 4.2. Salinity Differently Impacts the Bioavailability of Different Heavy Metal as Well as As and It Reduces the Heavy Metal Percolation in Soil Cultivated with K. pentacarpos

Although the bioavailability of meta(loid)s is not considered in a nutrient solution (even if it could be, to some extent, modified by speciation), it is a critical factor in polluted soil and directly influences the microbial community and the plant roots system as well as the entire soil ecosystem [83]. The bioavailability of As and three kinds of heavy metals (Cd, Pb and Zn) over the course of 3 months of *K. pentacarpos* cultivation was slightly different compared to 6-month trials using *K. pentacarpos*. Salinity increased Cd bioavailability in polluted soil in small columns after 3 months of cultivation, while it decreased Cd bioavailability in big columns after 6 months of cultivation. For the other heavy metals and As, it showed the same tendency: NaCl reduced As bioavailability and had a limited (Zn) or no (Pb) influence on the bioavailability of other elements. This aspect illustrated that bioavailability is not a fixed property but that it varies depending on the interaction between the root system and the surrounding soil [84,85]. Accordingly, the interaction between the solid substrate and the root system should be more deeply analyzed in the future in *K. pentacarpos,* but a sound approach will require an adapted specific experimental system.

Arsenic and heavy metal bioavailability had a high correlation with the accumulation of all of the pollutants in K. pentacarpos [64]. In Vromman’s study [86], salinity strongly increased As translocation from the root to the shoot of Atriplex atacamensis Phil. Although salinity impacts the bioavailability of Cd differently when exposed on a short or long terms basis, salinity has a positive influence on Cd accumulation in plant tissue on the whole [14]. It is an interesting point that in nutrient solution, NaCl always decreased Cd absorption and translocation to protect plants from heavy metal toxicity [14,31], while in soil, in the presence of salinity, the plants accumulated Cd [64], illustrating that data may strongly differ between hydroponic and soil approaches.

One possible theory to explain these observations is that Cd absorption is hampered in nutrient solution, which is the result of the formation of chlorocomplexes (mainly CdCl^+^) [87], while in contrast, the increasing Cd solubility induced by NaCl leads to accumulation enhancement in soil substrate [64]. The presence of an excess of Na^+^ could indeed help to detach the Cd^2+^ that is retained at the surface of clay particles, thus inducing an increase in Cd bioavailability [88]. In this case, Cd bioavailability has a critical effect on absorption and accumulation in halophyte species. In a long-term experiment, however, it was shown that salinity decreased rather than increased Cd bioavailability, which could have been due to the soil properties (low level of clay and high level of silt). For Zn, the accumulation concentration in plants growing on polluted soil was lower than it was in solution [64]. This could be attributed to different Zn solubility in soil/solution substrate. Salinity had a limited impact on Zn bioavailability [89], while it induced a decrease in the accumulation on both short- or long- term time basis. Although the Pb accumulation in *K. pentacarpos* is approximately 10 mg/kg dry leaf or stem material [64], such a low concentration should be regarded as the ultimate consequence of a low bioavailability of Pb in polluted conditions. However, considering the high biomass production of *K. pentacarpos*, it is still believed that it is able to absorb and accumulate this heavy metal, especially if adapted strategies are used to increase the bioavailability without increasing leaching processes [64].

*K. pentacarpos* reduced pollutant leaching [64], meanwhile, the additional salinity also strongly reduced pollutant leaching. It was verified that there was an abundant hydroxyl-group-rich mucilage on the surface of a developed *K. pentacarpos* root [28,30] that was able to store huge reserves of water. Another interesting point is that in columns with plants, salinity reduced the total amount of As that was able to be removed from the column system, while also slightly increasing the amounts of Cd, Zn, and Pb [64]. Most of the extracted As was removed by the leaching process in the presence or absence of NaCl. This phenomenon may depend on the properties of the considered element as well as the plant species. Overall, in long term, the growth and reproductive cycle of *K. pentacarpos* was disturbed after heavy metal and As accumulation in the plants, which caused the plants to enter the flowering stage in advance. This is thought to be one strategy that can be used to cope with pollutant constraints.

## 5. Conclusions

In summary, *K. pentacarpos* is not only expected to have economic value, but it is also considered to be a potential candidate that is able to cope with environmental constrains such as heavy metal stress due to its relative tolerance to metal toxicity. However, research on phytoremediation with *K. pentacarpos* is only in its initial phases. The mechanism (physiological, biochemical and genetic) that is able to manage with the relationship between salt absorption and meta(loid) accumulation in *K. pentacarpos* should be taken into consideration and determined clearly and in detail. In addition, the efficiency of using a halophyte to remove pollutants from contaminated saline is relatively low compared to hyperaccumulating plant species. Hence, genetic engineering approaches to develop transgenic plants with the characteristics of high biomass production, more metal translocation and accumulation ability, metal toxicity tolerance might be more beneficial in this respect.

## Figures and Tables

**Table 1 plants-10-02495-t001:** The main studies performed in *K. pentacarpos* in response to either single meta(loid) or multiple meta(loid)s under salinity.

	Meta(oid)s	Salt	Growth Conditions	Duration of Treatment	Highlights	Reference
Single meta(loid) treatment	Cu	50 mM NaCl	Nutrient solution *	3 weeks	Copper interferes K absorption NaCl alleviates the Cu toxicity	[13]
	Zn	50 mM NaCl	Nutrient solution	4 weeks	Zn induces a modification in the composition and structure of polysaccharides	[28]
	Cd or Zn	50 mM NaCl	Nutrient solution	4 weeks	NaCl differently interferes with Cd and Zn toxicities	[31]
	Cd	50 mM NaCl	Nutrient solution	4 weeks	NaCl affords protection against Cd by the improved management of oxidative stress and hormonal status	[43]
	Zn	50 mM NaCl	Nutrient solution	16 weeks	NaCl alters the Zn distribution in plants	[62]
	Zn	50 mM NaCl	Nutrient solution	16 weeks	The roots provides a valuable biological material for Zn retention	[63]
Multiple meta(loid)s treatment	Combination of Cd and Zn	50 mM NaCl	Nutrient solution	2 weeks	Plants have a special responses to combination of Cd and Zn toxicity	[14]
	Combination of Cd and Zn	50 mM NaCl	Nutrient solution	48 h	NaCl has an effect on osmolyte synthesis depends on considered metal and plant organs	[15]
	Combination of Cd and Zn	50 mM NaCl	Nutrient solution	3 weeks	NaCl influences the interactive effects of Cd and Zn on ethylene and polyamine synthesis	[41]
	Combination of Cd, Zn, As and Pb	0.24% NaCl	Soil	12 weeks	NaCl modifies meta(loid)s and by the *K. pentacarpos* and pollutants leaching	[64]
	Combination of Zn and Cd	50 mM NaCl	Nutrient solution	3 weeks	CK assumes key function in meta(loid)s resistance but its efficiency is lower in the presence of NaCl	[65]

* “Nutrient solution” is half-strength modified Hoagland nutrient solution.

## Data Availability

Not applicable.

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
