# Peer review of "Kosteletzkya pentacarpos: A Potential Halophyte Candidate for Phytoremediation in the Meta(loid)s Polluted Saline Soils"

_plants, 2021, doi:10.3390/plants10112495_

Round 1
Reviewer 1 Report
Dear authors vey nice work. A few spelling mistakes e.g. line 49 phtoremediation...
Also reference Tripathi et al is in the text but without number and there is not at the back. All references after the number 37 have to change
Reviewer 2 Report
TITLE:
Please standardize the use of heavy metal, metal or metal(oid)s throughout the manuscript. I like the term metal(oid) when metals and metalloids are involved. I do not understand the term “heavy metals polluted saline” saline soils? Salty soils?
ABSTRACT:
Line 14: delete “species” or conserve “halophyte species”. But “haloPHYTE plant” definitely not
Line 15: delete “is able”
Line 15: a quantitative data for “high amount of salt” is necessary
Line 18: change “more and more” heavy matel(loid), for quantitative data. Also change to metal(loid)s
Line 18: I do not understand what “which is posing” means in this sentence
Line 19: does “halophyte species” refer to k. Pentacarpos?
Line 20: please rewrite this sentence: “not only one heavy metals stress but also multiple mixed heavy metal toxicities which is more closed to the nature was researched”.
Line 21: this review
Line 22: i will change the word “deciphering” as a review summarizes the main studies already published
Line 23: please rewrite this sentence: “how the salt accumulated by plants impacts their behaviors to tolerate the heavy metals is discussed”. This review analyzed the response of many plants? Behavior? Heavy metals?
Line 25: candidate for what? Phytoremediation? Phytostabilization? Please specify
KEYWORDS
Almost all keyword are already included in the title. Please modify
SECTION 1
Line 32:i do not feel this sentence “For the botanical naming of the seashore mallow, it is indicated that this species should be named Kosteletzkya pentacarpos (L.) Ledebour and Kosteletzkya virginica (L.) Presl. ex Gray has to be relegated to synonymy according to the international Code of Botanical Nomenclature’s rule[1]” necessary for the manuscript as already mentioned before (line 30) and is ok
Line 36: which various?. Please be more specific
Line 37: K. pentacarpos
Line 37: Delete very well is so subjective
Line 37: the coastal area?
Line 37: define salt… which salt? Only one?
Line 38: delete “province as well as several other coastal provinces, including” and include “provinces” after Fujian
Line 41: ecological engineering for what?
Line 42: bioindicators? For what?
Line 44: add references. Delete “article”
Line 45 and 46: change “the salt” by “salts”. Change “mainly NaCl” by “i.e. NaCl” and add others. The same for metals
Line 47: please change this sentence: “to the complicated environment from the perspective of plant physiology and biochemistry. And it gives a in-depth perspective on how it will be utilized in phtoremediation in future”. I dislike the term “complicated environment” as well as “. And”
I will add a paragraph discussing about the main sources of metal(loid(s which contribute to pollute the coastal area and their main effect on biological conservation of coastal ecosystems
SECTION 2
Line 53 to 59: which plant responses? Obvious? There are strong asseverations that must be explained an appropriately cited.
Lines 60 to 71: several grammar errors that must be corrected. Mutual impact? First sentence confusing. Why R. communis? what is low? What is high? No matter? Short term combination? Zn affect Cd accumulation directly? Which transporters? References for these asseverations. Most of these aspect need to be better explained
Line 72 to 82: please rewrite this sentence “increase in hemicellulose content of in response to Cd as well as an increase in 77 mucilage which is mainly composed of pectin were record.” change the style of the citation It has been reported[9] XX et al, (9)
Lines 83 to 100: which compounds produced by plants are involved in metal sequestration to limit bioavailability in soil? Which is the threshold level? What about metallothioneins and other compounds? What is HM?
Lines 101 to 113: wang et al must be correctly cited wang et al (14). Cd+Zn? Meanwhile? Change many. Takahashi?? Fix the citation style see the author guidelines OsHMA2? Plant responses to metal stress often depend of the plant species. Maybe in rice but we do not know if k pentacarpos respond in the same way. This is a strong speculation
Lines 114 to 119: delete although. Change more and more. THAT should be. You have listed 3 issues not 2
In the section: why only these metals?? It must be clearly defined in the introduction
Lines 121 to 123: I dislike the question as a single paragraph. Contextualization is needed
Lines 124 to 132: reference for this: “Although Cd did not directly partici-125 pate in redox reaction, such as Fenton reaction, it is still capable to disturb the electronic 126 transmission chain in photosynthesis in K. pentacarpos.” are these Cd ad Zn metals found in natural environment or the soil were contaminated? What is NPQ?Which are the ratios of Cd ans Zn that inhibit the photosynthesis? Are these studies performed in saline soils? Which are the units for 690 and 730? Is the abbreviation Ap used later? Conclude the main idea of the paragraph
Lines 133 to 142: roots and leaves. From which reference are these data? Which species? What is MDA? Reference for this “For example, no difference was found between Cd 136 alone and Cd+Zn treatment for relative leakage ratio, which indicates that the membrane 137 was damaged mainly by Cd, resulting in an outflow of ions”. Reduce and rewrite the last sentence
Lines 153 to 163: how many is few hours? Add references to the sentences that are missing. Define NPT. Verify that abbreviations are defined previously (i.e. DHAR, GR, POX and others) if not add the full name and the abbreviation in ( ) the first time
Lines 166 to 180: which are the classical phytohormones? spermidine (Spd) and spermine 168 (Spd),? The same abbreviation? What is spm? References missing in several sentences
Lines 181 to 197: clearly specify if the studies are relate with K pentacarpos or not. What is PAs? Have been defined previously? Check the author guidelines for correct cite articles as author year. References missing in several sentences. You have defined spermidine as spd before.
Lines 198 to 220: define excessive. Verify if the abbreviation have been defined previously. References missing in some sentences
Lines 221 to 149: Ashraf and Foolad 2007 is not the reference style for MDPI journals and also for author year format. Check that abbreviations will be used further or if they have been defined previously. References missing in some sentences. Metal exposure... which metal? What experiment? Reduce the length of some sentence for more clarity.what is KvBADH
Lines 250 to 255: ok
Lines 256 to 260: water status, and plant nutrition were not discussed before
SECTION 3
There are extremely long title for each section.
Lines 265 to 275:delete very weel. Define nutrient solution. References missing in some sentences.
Lines 276to 283: Which transporters? Which salts? Nacl impact heavy metals? Really? References missing in some sentences.
Lines 284 to 307: References missing in some sentences. Specify in which plant species. What is “S”?. some sentences must be rewritten and all paragraph shortened
Lines 308 to 319: references missing in some sentences. Be more specific about metals.
Lines 322 to 333: correct the citation style according to the MDPI format. References missing in some sentences. Define moderate dose.
Is NaCl the only salt? There are no other salt in the saline soils?
Lines 334 to 346: change literatures. References missing.
Figure 1. I think that a better figure must be constructed
SECTION 4
Very long titles for the sections
Lines 356 to 360: check citations. References missing. Finish the paragraph with an idea,
Lines 361 to 366: in this manuscript, you have not studied this plant. What is T DM. delete “:” and rewrite the sentence. References missing in some sentences.
Lines 367 to 385: references missing in some sentences. Some sentences are confusing (two mentions to K pentacarpos in the same sentence).
Lines 402 to 413: check the MDPI citations style. References missing. Hydroponic?? Soils??? Which references?? Please clarify
Lines 414 to 432: which plant? References missing in several sentences. Some sentences must be rewritten (i.e. The 425 present work is also the first one to consider lead accumulation in K. pentacarpos). This is not a research study is a review.
Lines 433 to 444: this is not an study, is a literature review. No references in the paragraph.
Lines 445 to 456: I will add this paragraph as a conclusion and perspectives section.
A table summarizing the main studies performed in K. pentacarpos in response to multiple metal(loid)s under salinity must be included to the manuscript.
Reference list have several inconsistencies (full and abbreviated journal names)
Round 2
Reviewer 2 Report
Dear authors:
The manuscript has been considerably improved, although I believe that some of the comments I made in the previous review were not totally considered , especially in the final parts of the manuscript.
I recommend a last deep and slow reading to eliminate mistakes in writing, confusing sections and verify details such as abbreviations.
In the previous review I have made a lot of decomentaries with the aim of improving the quality of the manuscript. I hope they have considered them. These are my contributions.
change matel(loid) to meta(loid)s
line 29: delete the second "world"
line 33: delete heavy metal
line 55: put this information without ()
Line 88: delete it has been reported that
Line 123: what is mang?
Line 124: studies or the study?
Line 133: reference missing. Concentration?
Line 166: It still remaining abbreviations that are not un full
Line 168: if the abbreviations will not be further used, they are not necessary. Please verify all other abbreviations in the manuscript (POx and gr must be deleted)
Paragraph at line 175: Please define if these informations are of K. pentacarpos or not
Line 293: delete complicatedly
I feel the first sentence of the conclusion is confusing.
Author contribution statement missing
